Regulation of endoplasmic reticulum stress on autophagy and apoptosis of nucleus pulposus cells in intervertebral disc degeneration and its related mechanisms

Dai Jiuming 1
Liu Jin 1
Shen Yucheng 1
Zhang Bing 1 18643898096@163.com
Li Chaonian 2 chencnchaonian@126.com
Liu Zhidong 1
1 Department of Orthopedics, Binhai County People’s Hospital , Yancheng , China
2 Department of Traditional Chinese Medicine, Binhai County People’s Hospital , Yancheng , China
Haraguchi Tokuko
Electronic publication date: 2024 Apr 22
Publication date: 2024
Volume: 12
Electronic Location ID: e17212
Received 2023 Dec 7; Accepted 2024 Mar 19
Copyright: © 2024 Dai et al.
Copyright year: 2024
Copyright holder: Dai et al.
License: This is an open access article distributed under the terms of the Creative Commons Attribution License, which permits unrestricted use, distribution, reproduction and adaptation in any medium and for any purpose provided that it is properly attributed. For attribution, the original author(s), title, publication source (PeerJ) and either DOI or URL of the article must be cited.
License URL: https://creativecommons.org/licenses/by/4.0/

Keywords: Intervertebral disc degeneration, Endoplasmic reticulum stress, Apoptosis, Autophagy

Funding: The authors received no funding for this work.

==============================
Intervertebral disc degeneration (IVDD) is a common and frequent disease in orthopedics, which seriously affects the quality of life of patients. Endoplasmic reticulum stress (ERS)-regulated autophagy and apoptosis play an important role in nucleus pulposus (NP) cells in IVDD. Hypoxia and serum deprivation were used to induce NP cells. Cell counting kit-8 (CCK-8) assay was used to detect cell activity and immunofluorescence (IF) was applied for the appraisement of glucose regulated protein 78 (GRP78) and green fluorescent protein (GFP)-light chain 3 (LC3). Cell apoptosis was detected by flow cytometry and the expression of LC3II/I was detected by western blot. NP cells under hypoxia and serum deprivation were induced by lipopolysaccharide (LPS), and intervened by ERS inhibitor (4-phenylbutyric acid, 4-PBA) and activator (Thapsigargin, TP). Then, above functional experiments were conducted again and western blot was employed for the evaluation of autophagy-, apoptosis and ERS-related proteins. Finally, NP cells under hypoxia and serum deprivation were stimulated by LPS and intervened using apoptosis inhibitor z-Val-Ala-DL-Asp-fluoromethyl ketone (Z-VAD-FMK) and autophagy inhibitor 3-methyladenine (3-MA). CCK-8 assay, IF, flow cytometry and western blot were performed again. Besides, the levels of inflammatory cytokines were measured with enzyme-linked immunosorbent assay (ELISA) and the protein expressions of programmed death markers were estimated with western blot. It showed that serum deprivation induces autophagy and apoptosis. ERS was significantly activated by LPS in hypoxic and serum deprivation environment, and autophagy and apoptosis were significantly promoted. Overall, ERS affects the occurrence and development of IVDD by regulating autophagy, apoptosis and other programmed death.

Background

Intervertebral disc degeneration (IVDD) is a common and frequent disease in orthopedics, which seriously affects the quality of life of patients. With the changes of modern working practices and aging population, the incidence of IVDD is on the rise (Kos, Gradisnik & Velnar, 2019). According to statistics, about 80% of people suffer from neck and shoulder pain or low back pain (Feng et al., 2013). Among the global population of more than 6 billion, the number of people with IVDD is as high as 900 million, with 200 million in China alone (Gupta et al., 2010). At present, the pathological mechanism of IVDD is not clear and not well studied. Therefore, it is of great significance to investigate the pathological process of IVDD.

Intervertebral disc nucleus pulposus (NP) cells reside in an avascular and hypoxic microenvironment (Madhu et al., 2020). The nutrient supply of intervertebral disc NP cells mainly depends on the diffusion of capillaries in the tissues of both endplates (Sakai & Grad, 2015). As a result, intervertebral disc NP cells are chronically hypoxic and have limited nutrient supply (Gonzalez Martinez et al., 2017). Therefore, the biological properties of NP cells under ischemic and hypoxic conditions and their regulatory mechanisms have long been of great concern. Unlike other cells, hypoxia and serum deprivation play a very important role in the energy metabolism, phenotype maintenance, extracellular matrix secretion, cell proliferation and survival of NP cells (Chen et al., 2015; Wang et al., 2017). Moreover, cultured NP cells can survive for a long time in a hypoxic environment (Li, Liang & Chen, 2013).

Autophagy is considered to be a self-protective response of cells to harmful stimuli such as nutrient deficiency and hypoxia (Klionsky et al., 2021). In the case of nutrient deficiency, cells overcome nutrient deficiency and promote cell survival by autophagy to obtain energy and reduce macromolecules in vivo and in their own organ tissues (Kim & Lee, 2014). At the same time, autophagy can selectively remove damaged or excessive oxygen free radicals, DNA, peroxisome, endoplasmic reticulum, mitochondria and other substances in cells, and reduce the accumulation of abnormal proteins and organelles to maintain cell homeostasis (Mizushima & Levine, 2020). However, numerous recent studies have found that autophagy plays a critical role in promoting cell death in various cells in response to certain stimuli (Duan et al., 2020; Kim, Cheon & Ko, 2020; Kim et al., 2018). Overactivation of autophagy, accompanied by overdegradation of intracellular organs and tissues by lysosomes, further promotes cell apoptosis or leads to autophagic cell death (type II programmed cell death) (Green & Llambi, 2015). A decrease in the number of cells is another main characteristic of intervertebral disc degeneration, mainly caused by programmed cell death or apoptosis (Fuchs & Steller, 2011). A higher rate of apoptosis has been observed in patients with intervertebral disc degeneration or in animal models (Gruber & Hanley, 1998; Hirata et al., 2014). Programmed apoptosis can be observed in NP cells of herniated intervertebral discs in humans and compressed NP cells in rats (Chen et al., 2017; Yurube et al., 2023). Inhibiting mTOR complex 1 (mTORC1) protects against inflammation-induced intervertebral disc cell apoptosis, aging, and extracellular matrix degradation by inducing autophagy and activating the Akt signaling network (Yurube et al., 2020). In addition, study has shown that autophagy occurs in the lumbar intervertebral disc tissues of rats, and that the level of autophagy in intervertebral disc cells gradually increases with the degree of intervertebral disc degeneration (Chen et al., 2019). However, it has also shown that autophagy level is significantly decreased in the NP tissues of patients with IVDD, and that the up-regulation of autophagy level can protect intervertebral disc from degeneration (Tang et al., 2019). Therefore, we speculated that different levels of autophagy had different effects on apoptosis or death, thus exerting different regulatory effects on NP cells.

Endoplasmic reticulum (ER) is an important membranous organelle for protein synthesis, folding and secretion in eukaryotic cells. When the folding function of ER protein is disturbed by endogenous or exogenous stimulation, a large number of unfolded or misfolded proteins accumulate in the ER lumen, and a series of subsequent reactions are called endoplasmic reticulum stress (ERS) (Oakes & Papa, 2015). At this point, the cell initiates the unfolded protein response (UPR) to remove unfolded proteins and restore ER homeostasis. When ERS continued, UPR was not sufficient to remove the accumulation of unfolded proteins and damaged organelles, and autophagy was activated (Qi & Chen, 2019). In addition, when ERS is too strong or lasts too long, the overactivated autophagy can eventually cause cell death. Studies showed that the deterioration of ERS can lead to a series of changes such as apoptosis and autophagy, thus leading to cell death (Wu et al., 2018; Yang et al., 2018). However, it has been suggested that the inhibition of ERS also increases cell death (Ariyasu, Yoshida & Hasegawa, 2017; Lin et al., 2021; Yuan et al., 2017). What then is the role of ERS level in the regulation of autophagy and apoptosis in ischemia-hypoxia-induced NP cells?

Collectively, this study was conducted to clarify the effects of ERS on autophagy and apoptosis of NP cells from the perspective of in vitro and in vivo, in the hope of providing a new therapeutic strategy for the treatment of IVDD.

Materials and Methods

Portions of this text were previously published as part of a preprint (https://www.researchsquare.com/article/rs-3148566/v1).

Cell culture

Primary rat nucleus pulposus (NP, cat. no. BH-Y420) cells that purchased from Bohui Biotechnology Co., Beijing, China were cultured in Dulbecco’s Modified Eagle Medium/Nutrient Mixture F-12 (DMEM/F12, Gibco, Waltham, MA, USA) containing 10% fetal bovine serum (FBS, Gibco, Waltham, MA, USA) with 5% CO2 at 37 °C. Hypoxic treatment conditions were 2% O2, 5% CO2 and 93% N2. The NP cells in hypoxia group (Hypo) were placed in a three-gas incubator for 24 h of hypoxia. In serum deprivation (SD) group, NP cells were treated with serum deprivation and cultured for 24 h. NP cells in Hypo+SD group were treated with serum deprivation and hypoxia for 24 h. NP cells in LPS group were treated with serum deprivation and hypoxia for 24 h, and then NP cells were induced by lipopolysaccharide (LPS, 1 μg/mL, MedChemExpress, Princeton, NJ, USA) for 24 h to construct the IVDD cell model (Zhou et al., 2022). In this article, ERS inhibitor (4-phenylbutyric acid, 4-PBA, MedChemExpress, Princeton, NJ, USA) and ERS activator (Thapsigargin, TP, MedChemExpress, Princeton, NJ, USA) were used to explore the mechanism. NP cells with 2–5 passages were used for the study according to the product instructions.

Cell counting kit-8 assay

NP cells were injected into 96-well plates (8 × 103cells/well) incubated for 24 h. Subsequently, 10 µL cell counting kit-8 (CCK-8) solution (Beyotime, Jiangsu, China) was added to each well and the cells were incubated for another 2 h. Then, cells were detected by enzyme-labelled instrument (Thermo Fisher Scientific, Waltham, MA, USA) with a wavelength at 450 nm. The cell survival rate was calculated as follows: cell survival rate = [(experimental OD - blank OD)/(control OD - blank OD)] × 100%.

Immunofluorescence

NP cells with indicated treatment were fixed with 4% paraformaldehyde for 10 min and permeabilized with 0.25% Triton X-100 for 10 min. After that, NP cells were then incubated with anti-glucose regulated protein 78 (GRP78; cat. no. ab21685; 1:200, Abcam, Cambridge, UK) and anti-GAPDH (cat. no. ab181602; 1:500, Abcam, Cambridge, UK) at 4°C overnight, following which was the incubation with appropriate secondary antibodies (Alexa Fluor 488; cat. no. ab150077; 1:200; Abcam, Cambridge, UK) for 1 h. The fluorescence images were acquired using a fluorescent microscope (Olympus FluoView™ FV1000, Tokyo, Japan) with an excitation wavelength of 488 nm and an emission wavelength of 519 nm. Positive cells in all fluorescence images were counted using ImageJ to calculate the positive rate.

Green fluorescent protein-light chain 3 immunofluorescence

Transfection of green fluorescent protein (GFP)-light chain 3 (LC3) was performed according to the manufacturer’s instructions (Genomeditech, Shanghai, China). NP cells with indicated treatment were fixed with 5% paraformaldehyde for 10 min at room temperature, permeabilized with 0.3% Triton-X for 5 min, and stained with 4′-6-diamidino-2-phenylindole (DAPI) for 3 min. Finally, NP cells that washed by double distilled water (ddH2O) were mounted on tissue slides with mounting medium (Thermo Fisher Scientific, Waltham, MA, USA). A fluorescence microscope (Nikon Corporation, Tokyo, Japan) was used to capture and analyze fluorescence images. Positive cells in all fluorescence images were counted using ImageJ to calculate the positive rate.

Flow cytometry

The Annexin V-fluorescein isothiocyanate (FITC) Apoptosis Detection Kit (Beyotime, Jiangsu, China) was used to quantify the apoptosis of NP cells. NP cells that washed with PBS were resuspended in binding buffer and then incubated with FITC-labeled Annexin V and propidium iodide (PI) in the dark for 15 min. Cells were analyzed using the FACScan flow cytometry (BD Biosciences, San Jose, CA, USA). According to the permeability of cell membranes to FITC and PI under different conditions, cells with mechanical damage are shown in the upper left quadrant as (FITC−/PI+), live cells are shown in the lower left quadrant as (FITC−/PI−), non-live cells are shown in the upper right quadrant, which are necrotic cells or late apoptotic cells, as (FITC+/PI+), and early apoptotic cells are shown in the lower right quadrant as (FITC+/PI-).

Western blot

NP cells were lysed in radioimmunoprecipitation assay (RIPA) Lysis Buffer (Beyotime, Jiangsu, China) on the ice to extract total proteins. The protein concentration was detected with a bicinchoninic acid (BCA) protein assay kit (Beyotime, Jiangsu, China). A total of 30 μg proteins were separated by 12% SDS polyacrylamide gel electrophoresis (SDS-PAGE), and then transferred to polyvinylidene fluoride (PVDF) membranes (Amersham Biosciences, Buckinghamshire, UK). Membranes, which were blocked with 5% bovine serum albumin (BSA), were incubated with primary antibodies specific to LC3 (cat. no. ab192890; 1/2,000; Abcam, Cambridge, UK), GRP78 (cat. no. ab108615; 1/1,000; Abcam, Cambridge, UK), C/EBP homologous protein (CHOP; cat. no. #AF5280; 1/1,000; Affinity Biosciences, Cincinnati, OH, USA), spliced X-box binding protein 1 (XBP-1s; cat. no. #AF5110; 1/1,000; Affinity Biosciences, Cincinnati, OH, USA), Bcl-2 associated X (Bax; cat. no. ab32503; 1/1,000; Abcam, Cambridge, UK), B cell lymphoma-2 (Bcl-2; cat. no. ab194583; 1/1,000; Abcam, Cambridge, UK), Cleaved-caspase3 (cat. no. #AF7022; 1/1,000; Affinity Biosciences, Cincinnati, OH, USA), Caspase3 (cat. no. ab184787; 1/2,000; Abcam, Cambridge, UK), Caspase12 (cat. no. ab315271; 1/1,000; Abcam, Cambridge, UK), Beclin-1 (cat. no. ab207612; 1/2,000; Abcam, Cambridge, UK), p62 (cat. no. ab109012; 1/10,000; Abcam, Cambridge, UK), receptor-interacting protein 1 (RIP1; cat. no. #AF7877; 1/1,000; Affinity Biosciences, Cincinnati, OH, USA) and receptor-interacting protein 3 (RIP3; cat. no. #AF7942; 1/1,000; Affinity Biosciences, Cincinnati, OH, USA) at 4 °C overnight. GAPDH is a feasible House-keeping gene as internal references in the disc cells (Yurube et al., 2011). Therefore, GAPDH (cat. no. ab181602; 1/10,000; Abcam, Cambridge, UK) was used for normalization. On the next day, the membranes were incubated with appropriate secondary antibodies (cat. no. ab6721; 1/2,000; Abcam, Cambridge, UK). Finally, the protein blots were visualized using the enhanced chemiluminescence (ECL) reagent (Beyotime, Jiangsu, China). ImageJ was used to analyze the gray value of the target protein and the GAPDH. Calculate the relative expression level of the protein using the formula: Relative expression level of protein = grayscale value of the target protein/grayscale value of the GAPDH.

Enzyme-linked immunosorbent assay

The production of interleukin-1beta (IL-1β), tumor necrosis factor-alpha (TNF-α), inducible nitric oxide synthase (iNOS) and interleukin-6 (IL-6) in the supernatant of NP cells were evaluated with ELISA kits (Abcam, Cambridge, UK) according to the manufacturer’s instructions. The absorbance values of each well were measured at 450 nm using an enzyme marker, and the average absorbance values of the standards and samples were calculated for each set of replicates. The concentration of the standard was taken as the horizontal coordinate and the A450 value as the vertical coordinate, and a smooth line was used to connect the coordinate points of each standard. The corresponding concentrations of the samples were calculated from the absorbance values of the samples and the standard curve.

Statistical analysis

All data that expressed as means ± standard deviation were analyzed with GraphPad Prism 5. All experiments were independently repeated in triplicate and all experimental data were biologically repeated in triplicate. Statistical comparisons between two groups were conducted by Two-Tailed Student’s t test and comparisons among three or more groups were conducted with one-way analysis of variance followed by the Tukey’s post hoc test. P < 0.05 was considered statistically significant.

Results

Serum deprivation induces autophagy and apoptosis and hypoxia down-regulates autophagy of NP cells

Firstly, the present study examined the effects of Hypo and SD induction on NP activity, autophagy, ERS and apoptosis. The CCK-8 results showed a significantly lower level of cell activity in the SD group as compared to the Normal group (Fig. 1A). And compared with the Normal group, SD treatment significantly increased the expression of ERS biomarker GRP78 (Fig. 1B) and the proportion of NP cell apoptosis (Fig. 1C). In addition, SD treatment significantly increased the expression of autophagy-related protein LC3-II/LC3-I but downregulated p62 (Figs. 1D and 1E). However, compared with the Normal group, Hypo treatment had no significantly effect on NP cell activity (Fig. 1A), GRP78 expression (Fig. 1B), apoptosis (Fig. 1C) and LC3-II/LC3-I expression, but significantly increased p62 expression (Figs. 1D and 1E). Compared with Hypo group, cell activity and p62 expression in Hypo + SD group was decreased, while the expression of GRP78, proportion of apoptosis, and the expression of LC3-II/LC3-I was increased. However, compared with the SD group, the activity and p62 expression of NP cells in the Hypo + SD group were increased, while the apoptosis rate, GRP78 expression and LC3 expression were decreased. Moreover, treatment with chloroquine further decreased LC3 expression and further increased p62 expression in NP cells compared to the Hypo + SD group (Figs. 1D and 1E). These results indicate that the autophagy level of normal cultured NP cells in vitro is low, and serum deprivation can further induce autophagy and apoptosis. Hypoxia alone could down-regulate the level of autophagy in NP cells.

Figure 1 Serum deprivation induces autophagy and apoptosis and hypoxia down-regulates autophagy of NP cells.

(A) CCK-8 detected the cell viability. (B) IF assay detected the expression of GRP78. (C) Cell apoptosis was detected by flow cytometry. (D) IF assay detected the expression of LC3. (E) Western blot detected the expressions of LC3II/I and Beclin-1. The quantitative values are expressed as mean ± SD from at least three independent experiments, *p < 0.05, **p < 0.01, ***p < 0.001.

Effect of ERS on autophagy and apoptosis in IVDD model cells

Inflammation is an important pathological mechanism of IVDD (Chen et al., 2022). Therefore, in this study, the model of IVDD in vitro was established by inducing NP cells with LPS under hypoxic and serum deprivation condition. The mechanism of ERS on autophagy and apoptosis in IVDD model cells was further investigated using the ERS inhibitor 4-PBA and activator TP, which inhibit or promote ERS by inhibiting histone deacetylase (HDAC) and microsomal Ca2+-ATPase, respectively. The results showed that 4-PBA (0–200 μM) and TP (0–1 μM) had no significant effect on the viability of NP cells (Figs. 2A and 2B). However, the cell activity (Fig. 2C) was significantly decreased compared with Control group, and the expression of GRP78 (Fig. 2D), the proportion of apoptosis (Fig. 2E), and the expression of LC3 (Fig. 2F) were significantly decreased. Compared with LPS group, LPS + 4-PBA treatment increased cell viability, decreased apoptosis level, and downregulated the expression of GRP78 and LC3. While LPS + TP treatment showed an opposite effect to LPS + 4-PBA treatment.

Figure 2 ERS activates and promotes autophagy and apoptosis in IVDD.

(A) CCK-8 detected the cell viability. (B). IF assay detected the expression of GRP78. (C) Cell apoptosis was detected by flow cytometry. (D) IF assay detected the expression of LC3. The quantitative values are expressed as mean ± SD from at least three independent experiments, *p < 0.05, **p < 0.01, ***p < 0.001.

Next, we examined the expression of autophagy-, apoptosis- and ERS-related proteins and the results as shown in Fig. 3. LPS induced an increase in the expression of ERS-related proteins GRP78, CHOP and XBP-1S (Fig. 3A), and apoptosis-related proteins Bax, Cleaved caspase3 and Caspase12, and a decrease in Bcl-2 expression (Fig. 3B). Also, compared with the Control group, the expressions of autophagy-related proteins LC3II/I and Beclin-1 were significantly increased by LPS stimulation (Fig. 3C). In comparison with LPS group, the expression of GRP78, CHOP, XBP-1S, Bax, Cleaved caspase3, Caspase12, LC3II/I and Beclin-1 were significantly decreased, and the expression of Bcl-2 was significantly increased in the LPS + 4-PBA group. In contrast, the expression trend of these proteins in the LPS + TP group was opposite to that in the LPS + 4-PBA group.

Figure 3 The expressions of ERS-, apoptosis- and autophagy-related proteins.

(A) Apoptosis- (B) and autophagy-related proteins (C) The quantitative values are expressed as mean ± SD from at least three independent experiments, *p < 0.05, **p < 0.01, ***p < 0.001.

Moreover, it was worthwhile to mention that ERS was significantly activated by LPS in hypoxic and serum deprivation environment, accompanied by increased autophagy and apoptosis. Inhibition of ERS has a relatively significant inhibitory effect on autophagy and apoptosis in NP cells induced by LPS in hypoxic and serum deprivation environment. However, the further promotion of autophagy by activating ERS is relatively limited. Thus, the next step will be to further observe the changes that occur when apoptosis and autophagy are directly inhibited.

Effects of inhibition of apoptosis and autophagy on ERS, apoptosis and autophagy in IVDD cell models

The apoptosis inhibitor z-Val-Ala-DL-Asp-fluoromethyl ketone (Z-VAD-FMK) and the autophagy inhibitor 3-methyladenine (3-MA) were used to observe the effects of inhibiting apoptosis and autophagy on LPS-treated NP cells. They inhibit apoptosis and autophagy by blocking Caspase and phosphoinositide 3-kinase (PI3K), respectively. Interestingly, the addition of apoptosis and autophagy inhibitors did not restore NP cell activity, and the addition of apoptosis inhibitors even exacerbated the impairment of cell activity (Fig. 4A). Results obtained from IF assay showed that the expression of GRP78 was increased after z-VAD-FMK or 3-MA administration compared with LPS group (Fig. 4B). In addition, apoptosis inhibitor z-VAD-FMK significantly decreased apoptosis (Fig. 4C) and increased LC3 expression (Fig. 4D), but autophagy inhibitor 3-MA significantly increased the cell apoptosis (Fig. 4C) and decreased LC3 expression (Fig. 4D).

Figure 4 ERS affects the occurrence and development of IVDD by regulating autophagy and apoptosis.

(A) CCK-8 detected the cell viability. (B) IF assay detected the expression of GRP78. (C) Cell apoptosis was detected by flow cytometry. (D) IF assay detected the expression of LC3. The quantitative values are expressed as mean ± SD from at least three independent experiments, *p < 0.05, **p < 0.01, ***p < 0.001.

Meanwhile, WB experiments showed similar results to those described above. Compared with LPS group, z-VAD-FMK or 3-MA treatment increased the expressions of GRP78, CHOP and XBP-1S (Fig. 5A). Besides, by contrast with the LPS group, z-VAD-FMK treatment decreased the expressions of Bax, Cleaved caspase3 and Cleaved caspase12 and increased Bcl-2 expression. Nevertheless, in LPS+3-MA group, the expressions of Bax, Cleaved caspase3 and Cleaved caspase12 were increased while Bcl-2 expression was decreased (Fig. 5B). As Fig. 5C demonstrated, z-VAD-FMK treatment increased the expressions of LC3II/I and Beclin-1 when compared with those in LPS group, while 3-MA treatment decreased the expressions of LC3II/I and Beclin-1. The results showed that apoptosis or autophagy of LPS-induced NP cells under hypoxic and serum deprivation condition were significantly inhibited. The aforementioned results suggest that the use of apoptosis or autophagy inhibitors decreased the levels of apoptosis or autophagy in LPS-induced NP cells. However, the cell viability and the degree of ERS were not restored. This suggests that there may be other cell death mechanisms involved in the role of ERS in the occurrence and development of IVDD.

Figure 5 The effect of apoptosis inhibitor and autophagy inhibitor on ERS-, apoptosis- and autophagy-related proteins.

(A) Apoptosis (B) and autophagy-related proteins. (C) The quantitative values are expressed as mean ± SD from at least three independent experiments, *p < 0.05, **p < 0.01, ***p < 0.001.

The role of ERS in the occurrence and development of IVDD is related to the regulation of necroptosis

In addition to the traditional form of cell death known as apoptosis, many new forms of cell death have been discovered in recent years such as programmed cell death that do not depend on Caspases (Liu & Lieberman, 2020). Inflammation is considered one of the main causes of IVDD (Chen et al., 2020), and necrotic apoptosis is viewed as a programmed cell death process with pro-inflammatory functions (Bertheloot, Latz & Franklin, 2021). Therefore, here we detected the expression of necroptosis related proteins. The ELISA results showed that the levels of IL-1β, TNF-α, iNOS and IL-6 were significantly increased in the LPS group when compared with the Control group. The expressions of these inflammatory cytokines were further increased after z-VAD-FMK or 3-MA administration compared with the LPS group (Fig. 6A). The WB results showed that the expressions of necroptosis related proteins receptor-interacting protein kinase 1 (RIP1) and RIP3 were significantly increased in LPS group compared with Control group. Compared with LPS group, the expressions of RIP1 and RIP3 were significantly increased after z-VAD-FMK or 3-MA administration (Fig. 6B). These results suggest that ERS can affect the occurrence and development of IVDD by regulating autophagy, apoptosis and other programmed death.

Figure 6 ERS affects the occurrence and development of IVDD by regulating programmed death.

(A) ELISA detected the levels of IL-1β, TNF-α, iNOS and IL-6. (B) Western blot was used to detect the expressions of RIP1 and RIP3. The quantitative values are expressed as mean ± SD from at least three independent experiments, *p < 0.05, **p < 0.01, ***p < 0.001.

Discussion

IVDD, which has complicated mechanism, is often accompanied by low back pain, acute lower limb nerve root type of adverse symptoms such as pain, seriously affecting the quality of life of IVDD patients (Fenn, Olby & Canine Spinal Cord Injury, 2020). A number of studies have shown that the decrease in NP cells is the direct cause of IVDD. With the excessive apoptosis of NP cells, the synthesis of extracellular matrix is reduced, leading to the destruction of normal intervertebral disc structure and physiological function, and further aggravating IVDD (Ding, Shao & Xiong, 2013; He et al., 2018; Jiang et al., 2014). In recent years, delaying IVDD by regulating the relationship between autophagy and apoptosis of NP cells has become a popular topic (Hong et al., 2020; Lin et al., 2020). The intervertebral disc is the largest vaseless tissue in the human body, so intervertebral disc cells are always in a state of limited nutrient supply. During IVDD, the nutrient supply of intervertebral disc tissue, including NP cells, becomes even more limited. A previous study showed that the continuous activation of autophagy in normal intervertebral disc cells can ensure the energy supply of intervertebral disc cells when serum deprivation (Chang et al., 2017). At the same time, autophagy can remove damaged and dysfunctional proteins in cells to maintain normal cell function and vitality. In the process of IVDD, autophagy level was significantly lower than that of normal intervertebral disc, while the level of apoptosis was significantly increased (He et al., 2021; Xie et al., 2019; Xu et al., 2020). In our experiment, serum deprivation was used to simulate the hypotrophic state of intervertebral disc in vivo to induce autophagy. We found that serum deprivation significantly reduced the activity of NP cells, promoted the expression of LC3 and LC3-II/I, and down-regulated the expression of p62 protein, which was consistent with the previous results (Yurube et al., 2019). Suggesting that NP cell activity is reduced under SD conditions and is accompanied by activation of autophagy. Meanwhile, it was found that the apoptosis of NP cells was significantly increased during serum deprivation period, which was consistent with the statement that the apoptosis of NP cells was increased during IVDD period. However, it was also found that hypoxic condition failed to induce NP cells autophagy. On the contrary, to a certain extent, hypoxic condition inhibited the expression of autophagy. The possible reason is that prolonged serum deprivation can cause excessive autophagy of NP cells, while hypoxia promotes the survival of NP cells under serum deprivation condition by preventing excessive autophagy of NP cells. Therefore, we believe that the regulation of autophagy by hypoxia contributes to the survival of NP cells in the serum deprivation environment, and plays an important protective role in the pathological process of IVDD.

After cells are stimulated by hypoxia and serum deprivation, ER regulates the homeostasis of cell microenvironment through ERS response, which is closely related to cell apoptosis, proliferation, autophagy and senescence (Chen et al., 2018). It has shown that the activation of ERS in the process of IVDD can increase the apoptosis of NP cells, promote the degradation of extracellular matrix, and positively regulate the autophagy of NP cells (Tu et al., 2018). However, some researchers have suggested that inhibiting ERS can also increase cell death (Ariyasu, Yoshida & Hasegawa, 2017; Chang et al., 2017; Yuan et al., 2017). Currently, the effect of changes in ERS level has not been explained in combination with the characteristics of NP cells themselves and the environment. In our experiment, we found that the expression of GRP78 was significantly increased under serum deprivation and hypoxia. Subsequently, ERS inhibitors and agonists were administered to further explore the regulatory mechanism of ERS on autophagy and apoptosis. The inflammatory response induced by inflammatory cytokines is one of the primary causes of IVDD (Chen et al., 2022). The use of chemical reagents or drugs to induce the early inflammatory response in lumbar spine pathology is a common method for creating IVDD models (Kim et al., 2013). LPS is a type of endotoxin that induces inflammation by acting on Toll-like receptors (TLR4) on the surface of host cell membrane (Wang et al., 2023). Treating NP cells with LPS can simulate IVDD (Zhang et al., 2019). In this study, we found that ERS was significantly activated by LPS in the serum deprivation and hypoxic environment, and autophagy and apoptosis were significantly increased in this state. The inhibition of ERS on autophagy was relatively significant, with decreased apoptosis and improved cell activity. However, the further promotion of autophagy by activation of ERS is limited.

Considering this, we then used autophagy inhibitors and apoptosis inhibitors to treat NP cells to further observe the changes of direct inhibition of apoptosis and autophagy. We found that the inhibition of apoptosis or autophagy resulted in a significant decrease in cell activity, suggesting that other programmed death may exist (e.g., programmed necrosis). Necroptosis is a form of cell death with morphologic changes similar to normal necrosis, but is highly regulated. Study has shown that in IVDD, necrotic cells account for 80% of the total cells (Buckwalter, 1995). ERS promoted programmed necrosis of NP cells under compression (Lin et al., 2021). In addition, programmed death involves TNF-α, RIP1, RIP3, and release of inflammatory mediators (Belizario, Vieira-Cordeiro & Enns, 2015). In our experiment, it was found that after LPS induction, the expressions of IL-1β, TNF-α, iNOS and IL-6 in NP cells were increased, as well as the expressions of RIP1 and RIP3. After inhibition of autophagy and apoptosis, the expressions of above inflammatory cytokines, RIP1 and RIP3 were further increased. These results suggest that ERS not only affects the occurrence and development of IVDD by regulating autophagy and apoptosis, but also regulates necroptosis in IVDD model cells.

Of course, this study has several limitations. Firstly, in this study, LPS was used in combination with various inhibitors or activators to treat NP cells. There may be interactions between these chemical agents in various signaling networks, so further mechanistic studies are needed. In addition, this study observed that the role of ERS in LPS-induced NP cells may also be related to necrotic apoptosis, as evidenced by the detection of necrotic apoptosis-related proteins. However, further exploration is needed to understand how necrotic apoptosis is involved in the role of ERS in IVDD, utilizing relevant inhibitors or activators. Furthermore, cell experiments cannot fully simulate the in vivo environment, especially because the growth environment of NP cells differs significantly from that of other cells. Therefore, further in vivo studies using animal experiments are needed to explore the role of ERS in IVDD.

In conclusion, it can be preliminarily concluded that the activation of ERS can regulate the occurrence and development of IVDD disease by regulating autophagy, apoptosis and other programmed death. We will further discuss this finding in future experiments.

Supplemental Information

Supplemental Information 1 Raw data of cell counting kit-8 (CCK-8) assay, flow cytometry, western blot assay, and enzyme-linked immunosorbent assay.

Supplemental Information 2 Original bands from western blot assay.

Supplemental Information 3 A visual presentation of the key research and findings of this study.

Additional Information and Declarations

Competing Interests

Author Contributions

Data Availability

The authors declare that they have no competing interests.

Jiuming Dai conceived and designed the experiments, analyzed the data, prepared figures and/or tables, authored or reviewed drafts of the article, and approved the final draft.

Jin Liu conceived and designed the experiments, analyzed the data, prepared figures and/or tables, authored or reviewed drafts of the article, and approved the final draft.

Yucheng Shen performed the experiments, analyzed the data, prepared figures and/or tables, and approved the final draft.

Bing Zhang conceived and designed the experiments, analyzed the data, authored or reviewed drafts of the article, and approved the final draft.

Chaonian Li conceived and designed the experiments, analyzed the data, authored or reviewed drafts of the article, and approved the final draft.

Zhidong Liu performed the experiments, prepared figures and/or tables, and approved the final draft.

The following information was supplied regarding data availability:

The raw data is available in the Supplemental File.

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
