# Peer review of "Regulation of endoplasmic reticulum stress on autophagy and apoptosis of nucleus pulposus cells in intervertebral disc degeneration and its related mechanisms"

_PeerJ, doi:10.7717/peerj.17212_

## Round 0.1 · original submission · Major Revisions

Comments from two reviewers have been returned. I recognize that reviewer 1's comments are important, and please make revisions according to these review comments. In particular, please respond to comments regarding experimental design or make revisions. I hope these comments will help improve this manuscript.

**Language Note:** The review process has identified that the English language must be improved. PeerJ can provide language editing services - please contact us at copyediting@peerj.com for pricing (be sure to provide your manuscript number and title). Alternatively, you should make your own arrangements to improve the language quality and provide details in your response letter. – PeerJ Staff

Reviewer 1 ·

Basic reporting

The manuscript, entitled “Regulation of endoplasmic reticulum stress on autophagy and apoptosis of nucleus pulposus cells in intervertebral disc degeneration and its related mechanisms” describes the involvement of endoplasmic reticulum (ER) stress, autophagy, apoptosis, and necroptosis in rat intervertebral disc nucleus pulposus (NP) cells. This is an only in-vitro study using commercialized cells which pharmacologically modulates intracellular signaling. The authors’ idea is interesting; however, the study design is largely preliminary. Moreover, further additional assays and evaluations are required in respective experiments. The reviewer’s suggestion includes the followings.

1. Please attach the page number to help the review process.

2. Abbreviations should be spelled out, even which is common, e.g. CCK-8, DAPI, LC3, ddH2O, FITC, PI, BCA, ELISA, TNF, iNOS, IL, RIPA, SDS, PVDF, BSA, GRP78, CHOP, XBP-1s, GAPDH, Bax, Bcl-2, z-Vad-FMK, 3-MA, and others.

3. In this manuscript, particularly in the results, more hypothesis and rationale-telling description is recommended. Overall rewriting is necessary.

4. In the introduction, the authors try to explain target cellular mechanisms of ER stress, autophagy, apoptosis, and necroptosis. In particular, apoptosis and necroptosis appeared in the results section suddenly; therefore, please briefly explain the cellular mechanisms in the introduction. Review papers focusing on these mechanisms in disc cells should be helpful (JOR Spine 3:e1082, 2020) (N Am Spine Soc J 14:100210, 2023).

5. In l. 82, “to investigate” is not the study objective. “To clarify” is recommended.

6. In ll. 94 to 95, what is hypoxic ischemia? What causes hypoxic ischemia?

Experimental design

7. Please explain the molecular mechanism and interfering point(s) of intracellular signaling of the applied agents including ER stress inhibitor, inducer, apoptosis inhibitor, and autophagy inhibitor in the materials and methods or more preferably in the results section. Then, as these drugs have cross-talks during multiple signaling networks, further mechanistic investigation should be warranted as the study limitation. In fact, this is the biggest limitation of this manuscript.

8. For the use of commercialized rat disc cell badges, is not the approval of the Animal Care and Use Committee (ACAUC) necessary at the authors’ institution(s)? This is a critical issue to be clarified. If so, how many badges and lot. Nos. of cells were purchased? As the used cells are not a cell line, only one is not acceptable.

9. In ll. 87 to 97, which passage of cells were used for evaluation (experiments)? It should also be addressed whether monolayer or three-dimensional culture system was applied.

10. In ll.88 to 90, generally, cells purchased could lose the specific phenotype because of multiple freezing and thawing. Please show the data of rat notochordal disc NP phenotype based on literature evidence (J Orthop Res 33:283-93, 2015).

11. In cell-counting kit-8 (CCK-8) assay, immunofluorescence, GFP-LC3 immunofluorescence, flow cytometry, Western blotting, and enzyme-linked immuno sorbent assay (ELISA) of the materials and methods, not only how to perform experiments but also how to analyze and interpret the data, to identify apoptotic cells in flow cytometry, and to perform semi-quantification in Western blotting should also be described.

12. In immunofluorescence and Western blotting, please list the name of target and loading control proteins. The information on the used antibodies is also necessary.

13. In Western blotting analysis, why did the authors use glyceraldehyde-3-phosphate dehydrogenase as a loading control? Please cite previous evidence to validate it for rat disc cells (J Orthop Res 29:1284-90, 2011).

14. In immunofluorescence, please perform densitometry analysis of protein expression and/or positive cell counting.

15. In statistical analysis, it should be described how many technical replicates and biological replicates were applied. Normality assumption and one-tailed/two-tailed should be described. Figure graphs shows multiple post-hoc comparisons, in which the Bonferroni test would not be applicable. Other post-hoc tests including the Tukey test may be better. Please discuss and revise this with statisticians at the authors’ institution(s).

Validity of the findings

16. In the results, each figure in each subheading is more understandable for the readers.

17. Findings of CCK-8 assay are compatible with results of a previous report (Eur Spine J 28:993-1004, 2019). Please discuss with it. This paper should also be helpful to understand how to monitor autophagic flux.

18. Please explain why GRP78 expression was examined.

19. In analysis of autophagic flux using IF, measurements of LC3 expression is not enough. Counting LC3 puncta is required. In addition, using Western blotting, LC3-II expression analysis with LC3 turnover assay using chloroquine or bafilomycin A1 is highly recommended.

20. To monitor autophagic flux, not only LC3 but also identify other markers such as p62/ sequestosome 1 expression should be warranted. At least, LC3 expression under the absence of lysosomal inhibitor only is not acceptable.

21. What is the clinical relevance of lipopolysaccharide (LPS) during intervertebral disc degeneration and/or degenerative disc disease?

22. Although an ER stress inhibitor 4-PBA and activator TP were used to rat disc NP cells, how did the authors determine the applied concentration of these drugs? The optimal, effective but not toxic concentration should be selected based on CCK-8 assay. Decreased cell viability by LPS in the figure 2A would come from the toxicity.

23. In the figure 4A, why did the administration of an apoptosis-inhibiting z-Vad-FMK decrease cell viability? Please explain the molecular mechanism.

24. In the figure legends, please show the presented data patterns (mean ± standard deviation), applied statistical tests, and analyzed sample numbers.

25. In all immunoblots, label the molecular weight of target proteins.

26. In all images, the scale bars are too small. Please attach the bigger scale bars and units.

27. “XBP-1s” is shown in the figures but “XBP-1S” is used in the text. Similarly, caspases are “caspase 3” in the text but “caspase3” in the figures. Please describe it in the consistent manner.

Additional comments

28. In l. 75, what is “UPS”?

29. In l. 181, “LPAS” would be a typographic error.

30. In the text and figures, provide a space between the number and unit, if necessary.

Reviewer 2 ·

Basic reporting

The manuscript by Jiuming and colleagues is well-written and presents a clear and concise overview of the research on the role of endoplasmic reticulum stress (ERS) in regulating autophagy and apoptosis of nucleus pulposus (NP) cells in intervertebral disc degeneration (IVDD). The authors have clearly stated the research question, conducted relevant experiments, and drawn reasonable conclusions. The manuscript is well-organized and easy to follow, but a few minor grammatical errors and awkward phrasings could be smoothed out.
Line 56: In vivo should be in italic font
Line 75: What is UPS?
Line 145: Spelling mistake for “Hyop”

Experimental design

In Materials and Method, the author should include a "Microscopy" section providing the experimental details like excitation and emission filter used, etc.

Validity of the findings

No Comment

Additional comments

A graphical abstract summarizing the study's key findings would enhance the manuscript.

---

## Round 0.2 · accepted · Accept

I confirm that the authors have addressed the reviewer’s comments. This manuscript is now ready for publication.

Reviewer 1 ·

Basic reporting

The authors have carefully addressed responses to the reviewer’s concerns, showing new information and figures. The authors have significantly improved the manuscript and strengthened the message which the authors aim to carry.

Experimental design

The authors did not perform additional experiments for the validation of commercially available rat disc nucleus pulposus cells used in this study. To clarify the phenotype of cells, measurements of aggrecan, collagen type II, brachyury, CD24, and others in protein expression are recommended based on the recommendation paper (Risbud MV, et al. J Orthop Res 33(3):283-93, 2015). Nevertheless, findings shown are consistent and reasonable. However, in the future, please keep it in mind to confirm the maintained phenotype when cells are analyzed.

Validity of the findings

The methods and results sections have been extensively revised. Additional data and graphs have also been presented in the figures. Revision has been conducted successfully.